# A Mathematical Framework for Enriching Human–Machine Interactions

**Andrée C. Ehresmann [1],\*, Mathias Béjean [2] and Jean-Paul Vanbremeersch [3]**

[1] LAMFA, Université de Picardie Jules Verne, 80039 Amiens, France
[2] IRG, Université Paris-Est-Créteil, 94010 Creteil, France
[3] EHPAD Saint-Joseph, 80330 Cagny, France
\* Correspondence: ehres@u-picardie.fr

**Abstract:** This paper presents a conceptual mathematical framework for developing rich human–machine interactions in order to improve decision-making in a social organisation, S. The idea is to model how S can create a "multi-level artificial cognitive system", called a data analyser (DA), to collaborate with humans in collecting and learning how to analyse data, to anticipate situations, and to develop new responses, thus improving decision-making. In this model, the DA is "processed" to not only gather data and extend existing knowledge, but also to learn how to act autonomously with its own specific procedures or even to create new ones. An application is given in cases where such rich human–machine interactions are expected to allow the DA+S partnership to acquire deep anticipation capabilities for possible future changes, e.g., to prevent risks or seize opportunities. The way the social organization S operates over time, including the construction of DA, is described using the conceptual framework comprising "memory evolutive systems" (MES), a mathematical theoretical approach introduced by Ehresmann and Vanbremeersch for evolutionary multi-scale, multi-agent and multi-temporality systems. This leads to the definition of a "data analyser–MES".

**Keywords:** human–machine interactions; memory evolutive systems; artificial intelligence; anticipation

## 1. Introduction

In most social organizations, human–machine interactions are nowadays commonly considered as ubiquitous, the classic "machine" going from a simple computer to "a machine which displays human-like capabilities such as reasoning, learning, planning and creativity" [1]. Still, if the increasing use of AI is supposed to make all of these human–machine interactions more efficient, the impact of AI will depend on what we mean by AI.

The history of AI is difficult to establish. Some authors date it back to the 1950s and attribute its paternity to Marvin Minsky—one of the fathers of computer science and cofounder in 1956 of the Artificial Intelligence Laboratory at MIT. Minsky's dissertation in 1954 was entitled "A Theory of Neural Analog Reinforcement Systems and Its Application to the Brain Model Problem" [2]. In the late 1960s, Minsky published a number of important well-known works on similar themes, in particular [3,4] and later [5,6].

Other people date "real" AI much later and consider Newell and Simon to be the introducers of "symbolic AI" in 1976, writing: "A physical symbol system has the necessary and sufficient means of general intelligent action." [7]. Historically, symbolism is indeed the first of the two different paradigmatic approaches to AI. It is based on the definition of an arbitrary set of symbols and semantic rules that manipulate the symbols.

Many more recent AI-based systems draw on a more bottom-up approach named "connectionism" which seeks to explain intellectual abilities by using artificial "neural networks" or "neural nets". Among the connectionists, there are a large number of authors dealing with complex neural systems and learning, such as Changeux, Dehaene, and Toulouse [8], who use a physically based approach, or Rumelhart, Hinton, and Williams [9].

Still, a certain number of authors do not recognize a deep gap between the symbolic and the connectionist paradigms, with each of them having their own strengths and weaknesses. This often leads to the idea of finding ways to combine the two paradigms. For instance, Minsky stated in [10]: "Which approach is best to pursue? This question itself is simply wrong. Each has virtues and deficiencies, and we need integrated systems that can exploit the advantages of both".

This general view has been developed further in more recent research works. For instance, Zhang, Zhu, and Su [11] propose three generations of AI, with symbolism as the first and connectionism as the second. Doing so, they call for a "third generation artificial intelligence by combining the current paradigms", for they believe that "AI cannot achieve human behaviours by relying on only one paradigm".

We defend this point of view, even if we do not agree with their tentative definition of the third generation using a vector spaces framework. Instead, the aim of this paper is to provide a conceptual mathematical framework, based on category theory [12], for a future third generation of AI, and to show that this makes it possible to create multi-level artificial cognitive systems allowing for richer human–machine interactions.

To this end, we first consider a social organisation S, such as a company, an educative or a health institution. While preserving a kind of permanence or collective identity, we assume that such an organisation evolves over time due to changes in the different natures of its composition (e.g., members who leave S), structure (e.g., reconfiguration of material resources of S) and functioning (e.g., new ways of working or communicating in S).

Then, we analyse the case of a close collaboration between S and an evolutionary high-tech machine named a data analyser (DA) that S develops over time. This DA is not only able to gather and memorize data about the environment, past and current activities and potential difficulties, but is also able to develop common deep strategies, for instance allowing the DA+S partnership to generate rich anticipatory assumptions for risk prevention.

Let us note the particularity of such a DA on an example where S is a healthcare service. In this case, medical robotics teams can propose robots to streamline workflows and offer value in many areas by accomplishing specific and more-or-less repetitive tasks.

Our aim for DA is much richer. Our objective is that it should, by itself, be able to detect some symptoms (after a thorough medical learning) and be trained to autonomously select appropriate responses to resolve a situation. Another role we assign to DA is to develop rich interactions with S that allow it to participate in decision-making meetings as an actor and as an observer capable of detecting and seeking to correct misunderstandings between participants.

To describe the functioning of S over time, including the construction of a DA, we draw on the "memory evolutive systems" (MES) concept introduced by Ehresmann and Vanbremeersch [13–15], a mathematical model for evolutionary and complex multi-scale, multi-agent and multi-temporality systems, such as biological, cognitive and social systems. This leads us to define a particular kind of MES, named DA–MES, by which we study how DA can improve human–machine decision-making.

As we also wish to consider decision-making about the future, we analyse how a particular DA–MES may become "Futures Literate" (FL), referring to Riel Miller's work on rich anticipatory processes and assumptions [16]. This makes it possible to show how, while relying on collective intelligence knowledge creation processes, one can model the development of richer human–machine interactions, including anticipation and risk prevention.

The paper is organised as follows: in Section 2, we recall the general notion of MES. In Section 3, we introduce a particular kind of MES, called a DA–MES, in which we model the construction of a "multi-level artificial cognitive system", called a data analyser (DA), which we then trained to develop rich human–machine interactions that allow for finer decision making, including anticipation. In Section 4, we provide ideas for potential applications.

## 2. Methodological Approach: Recalls on MES

Memory evolutive systems (MES) were introduced by Ehresmann and Vanbremeersch [13–15]. They consist of a mathematical approach based on category theory [12] coupled with dynamic systems and are used for the study of "complex" evolutionary and adaptive systems such as biological, social and cognitive systems.

Such systems have:

(i)    a tangled hierarchy of complexity levels with multifaceted components;

(ii)   a multi-agent, multi-temporal self-organisation with a network of local co-regulators, each operating at its own rhythm with the help of;

(iii)  a flexible memory allowing for self-repair and adaptation to changes.

MES do not describe the invariant structure of the system but give a "dynamic model" that evaluates the system in its becoming, with the variation of its components and their interrelations over time, and with some of these interrelations disappearing while new ones appear.

### 2.1. MES H Associated to a System S

This MES H has interacting components of different natures:

(i)    individuals and a hierarchy of groups of interacting individuals;

(ii)   material components such as artefacts, computers, machines, (in a DA–MES, this will include the data analyser (DA) to be implemented);

(iii)  memory components such as contextual data, conceptual and procedural knowledge, algorithms, various memories, and also unconscious or implicit knowledge such as pure practices, heuristics, values of different kinds, and even affects and emotions.

The components and the links through which they can communicate are dynamic entities whose successive states during their "life" can depend on some physical attributes (e.g., activity at a given time, propagation delay and strength of a link, . . . ).

The configuration of H at a time t consists of the states at t of the components and links between them that exist at that time. In the MES H, this is represented by a category Ht which has for objects the states of the components existing at t, and for morphisms the states of the links between them. Over time, there are two kinds of change for configurations:

(i)    dynamic changes of the state of the components and links existing at t, for instance imposed by energetic constraints;

(ii)   structural changes leading to the possible loss or addition of some components or links.

In H, the change of configuration, or 'transition' from t to $t' > t$ is modelled by a partial functor from Ht to Ht', defined on the components and links at t which still exist at t', which maps their state at t on their new state at t'. As shown by Figure 1, the different categories Ht and the transition functors connecting them form an evolutive system [13], consisting in a functor H: T -> ParCat from the time category T to the category ParCat of partial functors between (small) categories.

### 2.2. The Hierarchical Structure of H

The MES H has a compositional hierarchy. As illustrated by Figure 2, this means that its components are distributed into a finite number of complexity levels, verifying the condition: at a time t, an object C of the configuration category at t of level n + 1 'combines' a pattern P of interacting components of levels ≤ n so that C alone has the same operational role as P acting collectively.

In categorical terms, this means that C is the colimit (or inductive limit, Kan [17]) of P. Formally, a pattern (or diagram) P in a category consists of a family of objects Pi and some morphisms between them. A cone (modelling a collective link) from P to C is a family of morphisms si from Pi to C well correlated by the morphisms distinguished in P. The pattern P admits a colimit cP if there is a cone from P to cP through which any cone from P

to C′ factors uniquely. A hierarchical evolutive system is an evolutive system in which the configuration categories are hierarchical categories.

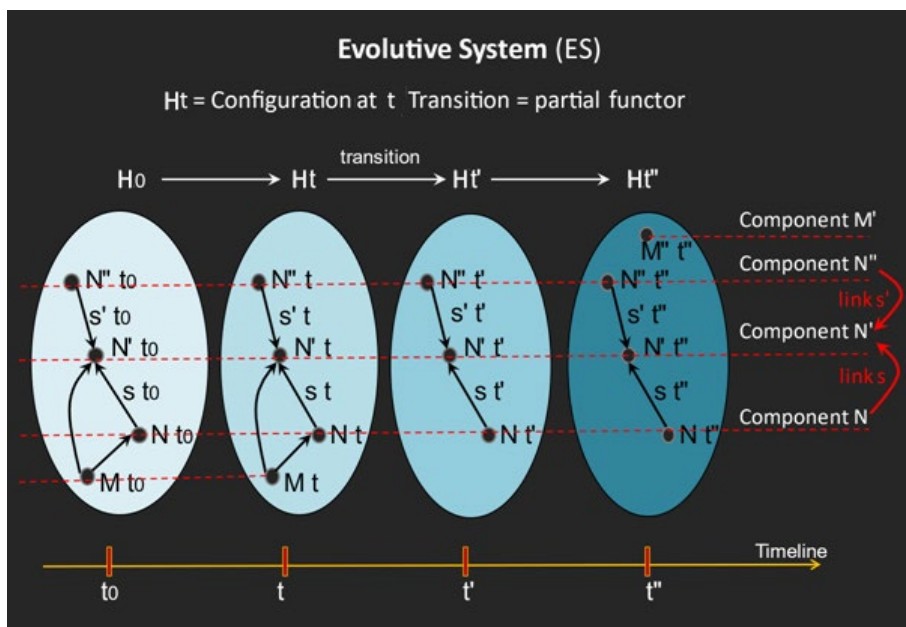

**Figure 1.** An evolutive system. This is defined by a functor from the category defining the order on the timeline to the category of partial functors between small categories.

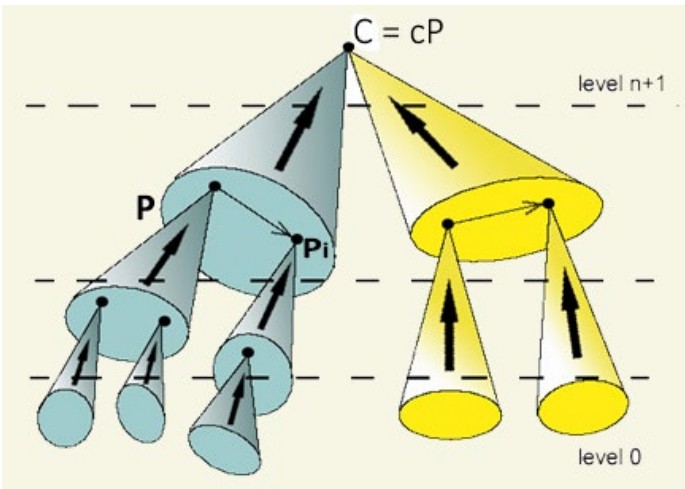

**Figure 2.** A hierarchical category. A category is hierarchical if the class of its objects is partitioned into a finite number of complexity levels, so that each object C of level n + 1 is the colimit of at least one pattern P with each Pi of level < n + 1; such a P is called a lower levels decomposition of C.

Over time, the decomposition P of C may vary progressively and eventually completely disappear while C persists. C acts as a "Janus": it is "simple" vs. higher levels, and "complex" vs. lower levels. Successive decompositions of C down to level 0 are named the ramifications of C. The order of complexity of C at t is the smallest length of a ramification of C; it is less or equal to the level of C. This definition is inspired by the Kolmogorov [18] and Chaitin [19] complexity of a string.

### 2.3. The Multiplicity Principle at the Basis of Flexibility

An important property of the MES H is the 'flexible redundancy' which generalizes the degeneracy property of biological systems (Edelman [20], Edelman and Gally [21]). This

asserts the existence of the components C which are multifaceted, in the sense that they can operate through (formally, are the colimit of) several structurally different non-connected lower-level decompositions and can switch between them; over time, they take their own individuation, independently of their lower-level constituents. In H, this property is called the multiplicity principle (MP). As shown by Figure 3, it allows for the existence, beside the simple links which bind clusters (Beurier [22]) of lower-level links, of complex links which are composites formed when the simple links bind non-adjacent clusters. In Chalmer's [23] terminology, these complex links are weakly emergent at their level with respect to the lower levels. The emergence of complex links is at the root of "change in the conditions of change" in Karl Popper's sense [24].

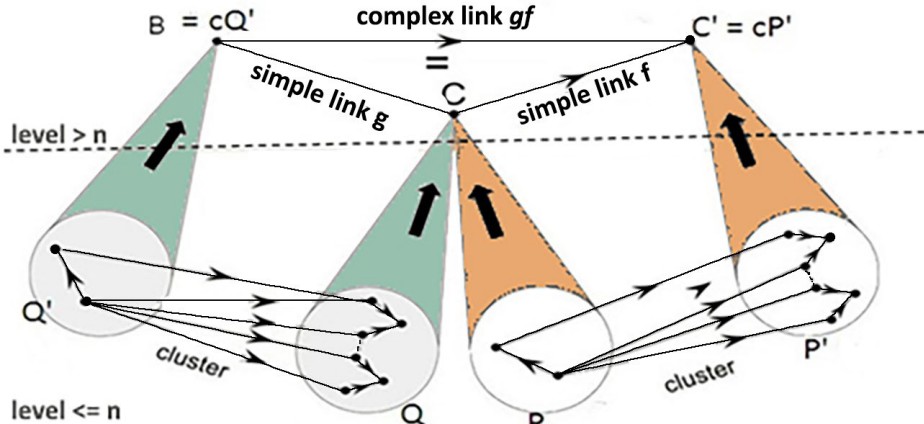

**Figure 3.** Simple and complex links. C is a multifaceted object colimit of both Q and P. The link g: cQ' -> cQ is a simple link binding a cluster G from Q' to Q, and idem for f binding a cluster from P to P'. Their composite is a complex link which is only weakly emergent at its level.

**Theorem 1.** *Complexity Theorem. MP is a necessary condition for the existence of components of complexity order > 1. Otherwise, we have pure reductionism. With MP we can speak of emergentist reductionism (in the sense of Mario Bunge [25]).*

MP gives flexibility to the system, in particular to the development of robustness though flexible memory (cf. Section 2.5) in which a component (named record) can be recalled through any of its different ramifications, providing plasticity over time to adapt to environmental changes.

### 2.4. The De/Complexification Process Leading to Emergence

As stated in Section 2.1, the change of configuration from t to t' is both due to dynamical changes of states and to structural changes. The structural changes correspond to the four "standard transformations" singled out by Thom [26]: Birth, Death, Scission, Confluence. In H, these correspond to the introduction of a new component (e.g., recruitment of a new employee), rejection of an existing one, and decomposition or formation of an interactive group of components. As illustrated by Figure 4, in the configuration category Ht, they correspond to the following operations: 'adding' external elements, 'suppressing' or 'decomposing' some components, and 'combining' a given pattern P into a new emerging component (due to become the colimit of P).

Given a procedure Pr with objectives of the above kinds on the category H, the de/complexification process for Pr consists in constructing a category H' in which these objectives are optimally satisfied (meaning H' is the solution of a universal problem). In [13], H' has been explicitly constructed and its 'categorical' construction gives conditions on Pr for the validity of the following:

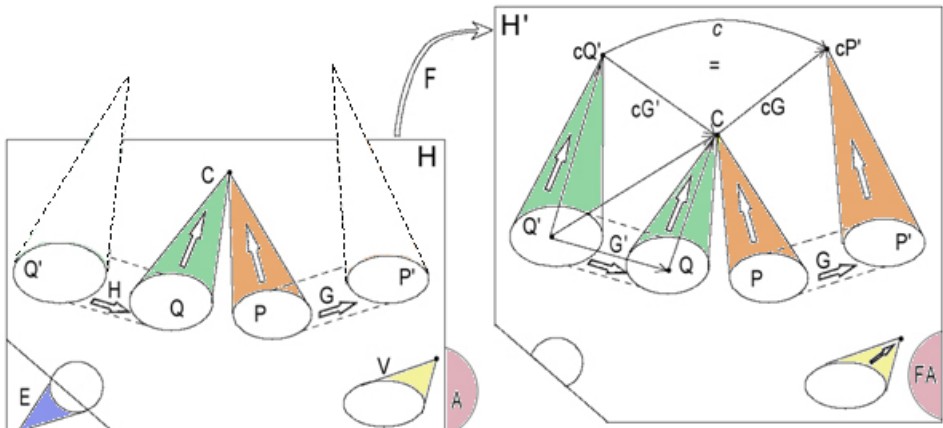

**Figure 4.** De/complexification process. We consider the procedure Pr on H with objectives: to add the set A, to suppress the cone E, and to add colimits cP′ and cQ′ to the diagrams P′ and Q′. The de/complexification H′ 'optimally' adds the colimit cones of bases Q′ and P′, the simple links cG and cG′ binding the clusters G and G′ and the complex link c: cQ′ -> cP′ which is their composite.

**Theorem 2.** *Emergence Theorem. For specific procedures, the de/complexification process leads to the emergence of (multifaceted) components of increasing complexity orders and of complex links, which render unpredictable the result of iterated de/complexifications.*

**Warning.** *The 'categorical' construction of H′ does not explicitly take into account the dynamic attributes of the components and links. For H′ to become the effective configuration of the system at a later time, it must be compatible with the different dynamic and physical constraints imposed by these attributes.*

*2.5. The Memory*

The social organization S develops a flexible long-term memory. In the MES H associated with S, the memory is modelled by a hierarchical evolutive subsystem whose components are named records. A multifaceted record takes its own individuation over time and can be recalled through its different ramifications. This memory develops over time through successive de/complexifications, and it therefore acquires records of increasing complexity orders (cf. emergence theorem which implies that iterated de/complexification processes give a categorical approach to "Deep Learning").

In the memory, we distinguish different types of records, among them:

(i)   A procedural memory whose records memorize some kind of action allowing certain objectives to be achieved; such a procedural record Pr is connected by 'command' links to a pattern of effectors through which it can be realized; for instance, a procedure for modifying the dynamic attributes of a component can be realized by an algorithm that computes the changes to be undertaken. If Pr has formerly been applied with success in a specific situation, it can exist an activator link from the record of the situation to Pr;

(ii)  A semantic memory which gradually develops through the classification of records into invariance classes represented by a specific concept [13].

*2.6. The Local and Global Dynamics*

H acts as a multi-agent self-organized system whose dynamics are modulated by the cooperation and/or competition between its processing agents. These agents, named co-regulators, can be simple individuals or formal groups of interacting people and/or machines (for instance, the data analyser of a DA–MES will act as a co-regulator). The overall dynamics weave the different internal local dynamics of the co-regulators. In the MES H, a co-regulator is modelled by an evolutive subsystem.

Each co-regulator CR has its own function, its differential access to the memory, in particular in recalling the procedures related to its function, and it acts stepwise at its own rhythm; the rhythms of the co-regulators can be very different.

A step of a co-regulator CR from t to t' is divided into three or more or less overlapping phases:

(i)     Formation of the landscape of CR at t, which collects the partial information of the system and its environment that is obtained via the active links arriving to objects of CR during the step from other parts of the system, e.g., the memory or other co-regulators. This information is analysed in order to make sense of it; (In H, the landscape is modelled by an evolutive system that has these active links for components.)

(ii)    Using the memory, a procedure Pr is selected through the landscape. It is not realized on the landscape but by activation of its commands to the effectors of Pr;

(iii)   This starts a dynamical process (eventually leading to differential equations) whose result will be evaluated at the beginning of the next step. If the objectives of Pr are not attained, in particular if Pr is not compatible with dynamic and temporal constraints, there is a fracture for CR.

At a given time, the various co-regulators may send conflicting commands to effectors. The global dynamic results form an 'interplay' among them, and this may cause a fracture to some of them. While the local dynamics can be computable, the interplay between co-regulators renders the global dynamic unpredictable.

### 2.7. MENS and Multi-Level Artificial Cognitive Systems

In *Entropy*, the paper entitled "MENS" [27] details the application of MES to the modelling of neuro-cognitive systems. The so-called model MENS is an MES whose level 0 of its hierarchy, NEUR, models the neural system whose configuration at a time t is the category of synaptic paths between neurons existing at t. The construction of its higher levels relies on two properties of the neural system (Edelman [20]):

(i)     The Hebb rule [28]: a mental object corresponds to the formation and reinforcement of a synchronous pattern of neurons;

(ii)    The degeneracy of the neural code, which will imply that MENS satisfies the "Multiplicity Principle" (MP, cf. Section 2.3).

Its successive levels, whose components are called cat-neurons (for category neurons), are constructed by successive de/complexifications of the level 0.

More precisely: as shown by Figure 5, a cat-neuron of level 1 models a mental object O which activates a synchronous pattern P of neurons as the "binding" (or colimit) cP of P added via a de/complexification of the category of neurons at t.

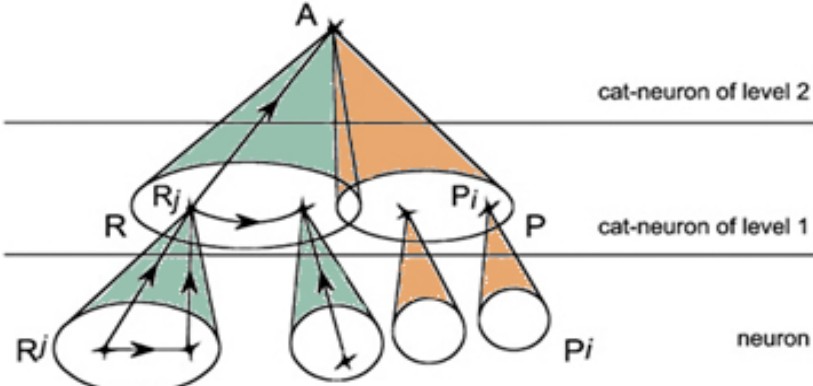

**Figure 5.** A cat-neuron of level 2. This is obtained via a double de/complexification of Neur and it gives a dynamic model of a mental object O by becoming the colimit of the different neural patterns activated by O.

Cat-neurons of higher levels are obtained by successive de/complexifications of lower levels so that a cat-neuron of level n + 1 is obtained by iterative binding of patterns of lower-level cat-neurons, which model flexible mental objects or processes of increasing complexity. Due to (ii) such a cat-neuron gives a "dynamic" model of a mental object O by becoming the colimit cP = cP′ in MENS of the various patterns P and P′ of the (cat-)neurons of lower levels able to activate O.

The number of levels increases over time, allowing for the formation of a robust and flexible memory with a higher level, called an archetypal core, driving the formation of conscious processes [13].

"**Multi-level artificial cognitive systems**", also defined in [27] (see the last section), are similarly defined as an MES obtained by successive de/complexifications of a category of paths of a graph satisfying the analogue of the Hebb rule and of the multiplicity principle. An important result obtained in [29] is that MENS supports both symbolism and connectionism (and even an "iterated connectionism" which defines a connectionism for each level). Due to the similarity of the constructions of the artificial cognitive systems, the same result is valid for each of them.

Now, if we accept the classification of AI in [11], it follows that such systems are of the third generation AI. If such a result seems more easily obtained here than in [11], it is because we use stronger mathematical tools based on category theory. For instance, the complexification process, in presence of the multiplicity principle, leads to the formation of multifaceted objects and complex links between them. Such an approach provides a way of combining both the connectionist and symbolist views, while the use of the vector spaces framework in [11] raises issues at each step.

## 3. Results: Constructing a DA–MES for Enriching Human–Machine Interactions

### 3.1. Definitions

Here, we define the concept of a DA–MES to model a continuous situation in which a social organization S collectively develops an internal multi-level artificial cognitive system (cf. Section 2.7), called a "**data analyser**" (DA). This DA should be able to collect and memorize a large number of different data and knowledge and, through rich human–machine interactions, act as a partner in collaborative decision-making.

As illustrated by Figure 6, the DA is conceptually equipped with four main units:

(i)     A "receptors unit" with different kinds of 'receptors' (e.g., sensors, user interfaces, etc.) and 'effectors' to communicate both ways between the system and its environment. Let us note that, unlike in [30], we do not seek to describe a specific implementation of this unit but only to indicate its role;

(ii)    A central "processing unit" to analyse and treat information, for instance by constructing de/complexifications;

(iii)   A "multi-level memory" which develops over time;

(iv)   An "output unit" which transmits commands to effectors.

Let us note that we are not looking for an explicit implementation of these units. We are only looking for a conceptual way for them to communicate, not an explicit practical way to implement them.

DA is "evolutive" in the sense that, in time, S may improve DA performances by configuring relevant changes (in hardware or software) to address current challenges effectively, for instance by increasing the number, the precision and/or the capacities of the receptors and effectors.

In the MES representing S, DA acts as a co-regulator of the system (cf. Section 2.6) able to accomplish the following operations, either alone or through interactions with higher co-regulators to form a collaborative work system:

(i)     As a CR, DA forms its landscape by continuously collecting material and behavioural data through its receptors (e.g., sensors) that come from the system and its environment, in particular from different co-regulators. As shown by Figure 7, it selects an

admissible procedure Pr with the help of the memory (seen as pr in the landscape) and sends its commands to effectors E. The result is evaluated at the end of the step,

(ii) DA helps to develop the memory. As shown by Figure 8, the activation of a pattern P of receptors is transported, via the landscape, into the activation of a pattern P′ in the processing unit. The record of P will be a colimit of P′ added via a de/complexification process.

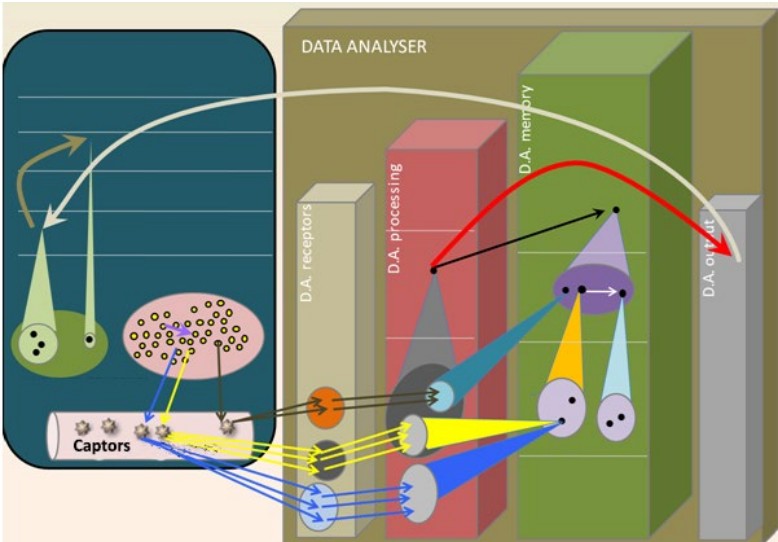

**Figure 6.** Presentation of DA. A DA with its 4 main units and the different kinds of interactions it has with components of the system S.

This memory is organized in a "relational database" by evolutionary computing. DA also cooperates with higher co-regulators to develop the global memory of the system S and organize it into multiple levels up to the formation of a conceptual level (in the semantic memory) and of more complex procedures and procepts (in the procedural memory).

(iii) Another important role of DA will be in helping in decision-making by interacting with higher co-regulators. Thus, the MES also acts as a "collaborative decision-making system" [31]; however, eventually DA can itself select already known procedures and realize them (by activation of their effectors).

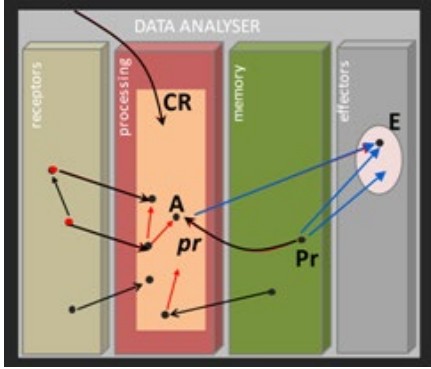

**Figure 7.** DA landscape. DA acts as a CR by steps. One step is divided into three parts: (i) formation of the landscape which is an ES with components that are the links activating at least one element of DA during the step; (ii) selection of a procedure Pr in the memory via pr; and (iii) sending the commands of Pr to its effectors. There is a fracture if the results of Pr are not achieved at the end of the step.

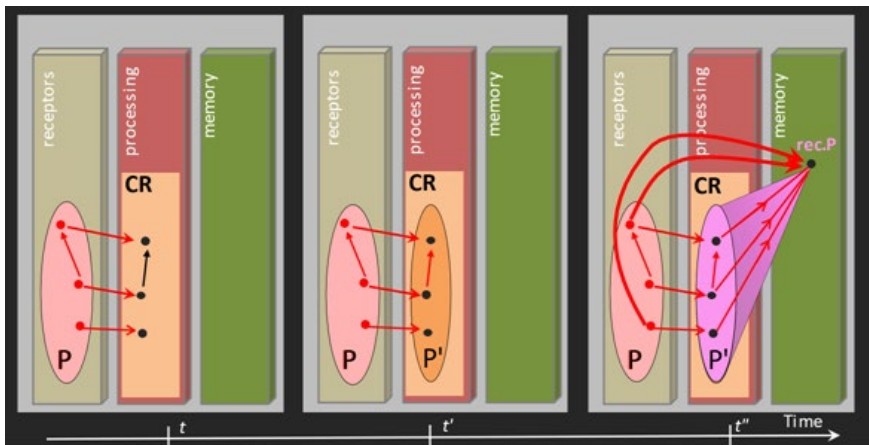

**Figure 8.** Formation of a record in DA memory. The activation of a pattern P of receptors leads (via the landscape) to the activation of a pattern P′ in the processing unit. The record recP of P is obtained as a colimit of P′ added via a de/complexification process.

### 3.2. General Functioning of DA–MES

To simplify, when no confusion can arise, we will not make an explicit distinction between a social organization S equipped with a data analyser DA and its associated DA–MES in which DA operates either by itself or in coordination with humans.

By itself, DA acts as a co-regulator of the MES: at a given time, its landscape gathers the information received from its receptors, coming from the memory or sent by other co-regulators. Depending on the number and precision of its receptors, it can distinguish some weak signals (e.g., a small anomaly in the data sent by a sensor) and alert the system. If it has already met a similar experience, it can even select (through an activator link) a procedure used to correct it and activate its effectors. This allows for a quicker answer, possibly avoiding risks that are more serious, but may need some control to avoid errors or unethical behaviour.

However, we are going to show that the main role of DA is to cooperate with a human co-regulator G to ensure high quality decision-making, with help from the development of rich human–machine interactions. In this situation, the two co-regulators G and DA and the different links which connect them act as a (macro-)coregulator of the DA–MES. Its landscape, named the macro-landscape (ML), constitutes a collective working space in which we are going to show that G and DA proceed as follows:

(i)  They share information and knowledge of different kinds, thus forming a pattern AG called the pattern of 'G-archetypal' records;
(ii)  They construct the macro-landscape ML with the help of AG and analyse it to make sense of the present situation (retrospection process);
(iii)  They search for adequate procedures to answer the situation (prospection process) and finally reach a consensus decision.

In these operations, DA does not just operate as a multi-level database but also as an information collector able to detect some weak signals, and as an active coordinator.

### 3.3. Formation of a G-Archetypal Pattern AG of Shared Records

Initially, the members of G have different individual expertise and knowledge. A higher order multifaceted record C, such as a polysemic concept integrating knowledge of different modalities (explicit or not), may have different meanings for two members of G depending on the ramifications through which they recall it. Exchanges of information between them to reach a common understanding are perceived by DA receptors and memorized so that DA may store C with all of its different ramifications, whence forming a common enriched perspective C* of C accessible to all. C* encompasses the different

meanings of C and eventually some tacit knowledge, such as emotions aroused in the course of the discussion. C* is called a G-archetypal record.

As illustrated by Figure 9, the G-archetypal records are connected by loops of strong and fast complex links, which self-maintain their activity for a long time. With these links they form the evolutive G-archetypal pattern, AG. (The development of AG is a consequence of the emergence theorem).

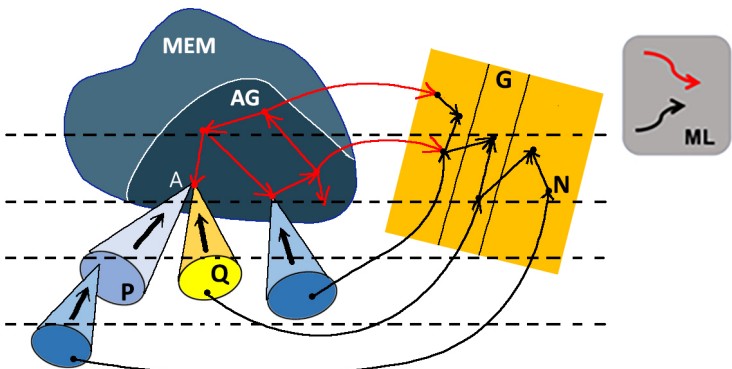

**Figure 9.** Improving decision-making. Let G be a group of humans acting as a co-regulator of S. By working with G, DA allows a better cooperation between persons in forming a G-archetypal pattern AG of shared concepts. This allows the formation of a common macro-landscape, ML, in which retrospection and prospection processes can develop.

**Remark 1.** *The formation of AG will be improved if DA can record voices through hearing devices, memorize them, and process them to infer some personality traits of the speakers such as dominance or trustworthiness (Ponsot et al. [32]). It may also detect some primary emotions (e.g., pleasure, or arousal) from their attitudes and deduce other emotions (using the computational PAD Emotion Model, Zangh et al. [33]). This would provide DA with a kind of theory of mind (ability to impute unobservable mental states to others); for M. Kosinski [34]: "this ToM-like activity (thus far considered to be uniquely human) may have spontaneously emerged as a by product of language models' improving language skills".*

*3.4. Construction and Analysis of the Macro-Landscape ML*

AG acts as an engine for the construction of the macro-landscape ML which contains the landscapes of G and DA, interconnects them and extends them both spatially and temporally Indeed, the recall by G or DA of a G-archetypal record C* in their landscape first diffuses in AG through archetypal loops, then propagates to lower levels through the unfolding of ramifications and switches between them, thus extending ML to lower levels. Moreover, ML lasts longer due to the self-maintained activation induced by AG.

In ML, current observations and recent events can be related to past, more-or-less similar cases, allowing to sense and to make sense of the present situation, its trends and its possible evolution. This retrospection process is followed by a prospection process, still using the engine role of AG, to search for possible anticipatory assumptions, to 'virtually' evaluate their risk of dysfunction, and finally to select one. Once a consensus decision has been taken, it is memorized by DA together with the rejected dissenting views, as well as its later outcomes, to be of help if a similar situation recurs.

**Remark 2.** *The above constructions generalize those undertaken in a D–MES (Béjean and Ehresmann [35]) to model how a simple group of humans collaborates. The benefits of introducing DA are the detection of more (even weak) signals and the development of a more efficient collaboration thanks to a larger sharing of explicit or tacit knowledge.*

## 4. Applications and Conclusions

### 4.1. DA–MES Becoming Futures Literate (FL) [16,36]

For a system to become FL means that it develops a rich variety of anticipatory assumptions (AA) which "are the fundamental descriptive and analytical building blocks for understanding FL" and "using-the-future" (Miller [16], p. 24). An AA for a co-regulator CR (at a given time t) corresponds to the choice of a specific procedure Pr on its landscape and evaluation of its expected result (if the realization of Pr succeeds). These AAs can be explicit (conscious or not for human groups) or only tacit. The rich human–machine interactions with DA can help transform tacit AAs into explicit ones that become realizable. For instance, the anticipation of a risk will help to prevent it.

In Miller [16] (Chapter 2), AAs are classified in different groups. Here we adapt this classification in our frame.

(i)     AAs corresponding to some kind of "repetition". These correspond to cases where the present situation has already been met, and there is an activator link from its record to a procedural record which had given a satisfying result. The AA will then consist of the activation of the same procedure (via the activator link). An example is given by the use of statistics, where this represents a kind of "colonization" of the future, impeding real novelty. In cases where the situation has prompted different responses in former occurrences, there can be several activator links, and some choice will have to be made. In any case, the result of the procedure can be different from the expected one, due to possibly unrecognized changes in the situation;

(ii)    AAs corresponding to novel futures in answer to the detection of "Specific-Unique" phenomena such as weak signals. These may require new procedures, eventually leading to the emergence of higher complex objects and links;

(iii)   In a DA–MES, the richer interactions between DA and G make it possible to find more AAs, e.g., corresponding to weak signals, and to answer with a richer stock of innovative procedures on extended landscapes. Thus, the system increases its futures literacy.

### 4.2. An Illustration in the Case of Risk Prevention

Let us give a potential illustration that has been at the root of this research work, namely the problem of health, or risk prevention in a care-home for elderly persons, developed in Ehresmann and Vanbremeersch [37].

In this scenario, the medical staff (physicians and nurses), S, trains an evolutionary multi-level artificial cognitive data analyser, DA, to assemble a large number of medical knowledge and personal data on the residents (collected through non-invasive devices), to analyse them, and to learn, from S, possible treatments and their effects. The DA–MES conceptual framework could show how, together, they can develop creative processes to monitor health risks and, as much as possible, prevent or reduce stressful events. Once DA has memorized pathological symptoms, it can quickly recognize them, inform the medical team and eventually begin an already used adequate treatment. Thus, pathologies are recognized and cured more quickly, e.g., preventing dissemination of epidemics.

### 4.3. Conclusions

The aim of this article was to study how a social organization of any kind, say S, can improve its internal cohesion and its external output by designing a high-tech multi-level artificial cognitive system, called a data analyser (DA), able to cooperate with certain human teams to improve their collaboration and so achieve a better quality of decisions.

Generally, in such a social organisation S, there are a number of high-tech machines, acting as lower-level specialized co-regulators, but these machines are usually controlled by human rules, leaving them with little freedom. In this paper, the idea was different because we aimed to consider a rich collaboration between a multi-level artificial cognitive data analyser, DA, constructed to act on its own and anticipate future situations.

To do so, we used the MES (Ehresmann and Vanbremeersch [13]) framework, a conceptual mathematical approach based on category theory, developed for studying complex evolutionary hierarchical systems, such as social or cognitive systems. The global dynamic of such systems is obtained by an interplay between the dynamics of a family of sub-systems, called co-regulators, acting as agents, each of which has its own complexity level and temporality (cf. Section 2).

We then introduced the notion of a DA–MES (in Section 3), namely an MES which, in time, modelled the building of a customized evolutionary multi-level artificial cognitive DA with specific characteristics to make it more autonomous and independent from humans than usual machines, even high-tech machines, and to become more anticipatory over time. It was limited only in its assignment to respect human ethical rules.

The DA–MES framework opens new ways for modelling the collaboration between machine and human teams. In particular, it provides conceptual tools to study how the artificial cognitive system DA can collaborate with a human group working on a certain topic. By modelling the different devices of DA as well as the characteristics of the co-regulator G representing the human group (cf. Section 3), it is possible to formalize how DA can exchange both ways with the members of G, considering the case where DA explicitly participates in the discussions and even possibly supervises them. For instance, the DA–MES framework renders it possible to model the way in which two members of G can discuss the "same" multifaceted object by using its different facets, e.g., by using two meanings of the same word, without realizing the differences. In this situation, the DA–MES makes it possible to formalize at least two contrasting cases. First, the case where the discussion leads to a misunderstanding threatening the cooperation between the two G members. Second, the case where DA identifies the problem with the multifaceted object, memorizes the two meanings by forming an archetypal object (cf. Section 3) and finally communicates with the two G members to facilitate their cooperation. The DA-MES could even consider the case where DA identifies the way in which some emotions make it difficult to reach a common understanding.

As a whole, the article provides a conceptual mathematical framework to conceive and design artificial cognitive data analysers (DA) (which, as mentioned at the end of Section 2, could exemplify a third generation AI) able to enhance the collaboration between humans and machines. While potentially improving the quality of the decisions made, such new DA would also be able not only to keep track of the decisions made, but also of the way they were made, so that they could be recalled in cases where a similar experience occurs later on. In particular, in Section 4, we gave an application in which the presence of such a DA can be useful for decisions about the future. This allows the DA–MES to develop better anticipatory assumptions, as far as becoming "futures literate" (as described in [16]). In practical situations, this may help prevent some risks and lead to better futures.

**Author Contributions:** Conceptualisation and methodology: A.C.E., M.B. and J.-P.V. have equally contributed to this paper. This is a consequence of the fact that they have been working together for many years in adjacent domains and is shown by the necessary self-references to some of their older publications in the Bibliography; Writing: A.C.E. and M.B.; Graphics: J.-P.V. All authors have read and agreed to the published version of the manuscript.

**Funding:** This research received no external funding.

**Data Availability Statement:** Not applicable.

**Conflicts of Interest:** The authors declare no conflict of interest.

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
