# Peer review of "A Mathematical Framework for Enriching Human–Machine Interactions"

_make, doi:10.3390/make5020034_

Round 1

Reviewer 1 Report

The paper presents a conceptual mathematical framework for enriching human-machine interactions to improve decision-making in a social organization. According to the Authors, the idea is to model how S can create a customized artefact, collaborate with humans in collecting data, learn, anticipate situations and develop new responses, thus improving decision-making. The topic is interesting and the paper corresponds to the journal's aim and scope.

However, there are shortcomings in this paper. In the Introduction, the Authors’ contributions should be emphasized. The considered problem should be clearly stated. There is also a lack of related work. Do similar solutions exist?

Moreover, the graphical part should support the proposed framework and be well described. It helps better understand the Authors’ idea. I suggest rebuilding the paper.

The list of references is relatively low.  It contains only 13 positions (5 of them are Ehresmann).

The Authors' contributions are not specified.

Author Response

We recognize that the first version of the paper (submitted February 2) which was examined by this Reviewer had a number of shortcomings which we tried to correct in a second version dated February 15, and still more in the revised version presented here. Let us be more precise.
1. "The considered problem should be clearly stated". At this end, we have much expanded the Introduction and the end of Section 2, as well as the beginning of Section 3 (implementation of the Data Analyzer)
2. "There is also a lack of related work. Do similar solutions exist? " We have not found papers proposing "related work" with "similar solutions" One reason could be that we essentially use constructions done in the Theory of Categories which does not seem used by authors on human-machine interactions.
3. "the graphical part should support the proposed framework and be well described. It helps better understand the Authors’ idea. I suggest rebuilding the paper". We have added such a Graphical Part consisting of 9 large enough Figures, distributed in the text, each one with an explicit legend describing what the figure is supposed to represent. We hope that the inclusion of this part makes it not necessary "rebuilding the paper".
4. "The list of references is relatively low. It contains only 13 positions". We have much expanded it since it now contains 37 references.
5. The Authors' contributions have been added.
+
A. Ehresmann, M. Béjean and J.-P. Vanbremeersch

Reviewer 2 Report

The mathematical framework proposed seems novel to be published. I recommend it can be published. 

Author Response

Though this report is already favorable, we would like to add that the new version of the paper (May 1st) has been thoroughly expanded. In particular:
1. Addition of an important Graphical Part including 9 Figures with developed legends,
2. Addition of references so that there are now 37 references.
3. The Introduction, the end of Section 2 and the beginning of Section 3 (implementation of the Data Analyzer) are now much more developed.
A. Ehresmann, M. Béjean and J.-P. Vanbremeersch

Reviewer 3 Report

The paper proposes a novel approach using a data-analyser (DA) as part of a Memory Evolutive System (MES) to capture rich human-machine interactions (HMI). The authors contend the DA-MES can capture contextual data and implicit knowledge done by humans within an organization over time and can assist in improved decision making. The authors have a long history of publishing in about MES, which gives their ideas credibility.

However, there are some areas where the paper could be improved. The authors do not provide enough detail about the specific algorithms and implementation of the DA-MES approach. Additionally, there are no results or analysis presented to support the claim that the DA-MES can actually capture contextual data and implicit knowledge involved in human cognition.

Moreover, the paper would benefit from the inclusion of figures and diagrams to better illustrate the ideas and concepts presented. The authors could also expand on the example scenario to provide more insight into how the DA-MES actually works and how it would be implemented between the CPU and memory. It is also unclear how the DA-MES approach would differ from current databases and what the authors consider a "rich" interaction.

Overall, the paper presents an interesting approach that has the potential to enable rich HMI. However, the paper lacks detail in some areas and would benefit from additional analysis and illustrations to support its claims.

Author Response

1. We are in full agreement with the Reviewer when he says that the paper "would do better illustrate the ideas and concepts presented" and (later) "benefit from additional analysis and illustrations to support its claims". For this purpose, in our present Revised paper we have introduced an important Graphical Part to make more accessible the meaning of concepts that play a central role, for instance the implementation of the Data Analyzer (DA) in Section 3. This graphical part consists of 9 large enough figures, distributed in the text (Sections 2 and 3), each one with an explicit legend describing what the figure seeks to represent.
2. "The authors do not provide enough detail about the specific algorithms and implementation of the DA-MES approach". The DA-MES approach does not rely on specific algorithms, but is based on more abstract categorical constructions which are made more understandable thanks to added specific figures. In particular, we contend that "the DA-MES can actually capture contextual data and implicit knowledge involved in human cognition" using the de/complexification process (Figure 4) at the basis of the formation of a memory (Figure 8) with more and more complex records (Emergence Theorem, cf. Section 2).
3. This also explains "how the DA-MES approach would differ from current databases" since our model is based on the Theory of Categories which does not seem being used by people working on Human-Machine interactions. It is also this framework which allows to define the concept of a "rich interaction" between DA and a human group G, thanks to the formation of a common extended 'data-base' called the G-archetypal pattern (Figure 9) which helps bringing together the points of view of both sides
4. "The authors could also expand on the example scenario to provide more insight into how the DA-MES actually works". We recognize that this would be interesting but expanding the example could raise Confidentiality issues.
A. Ehresmann, M. Béjean and J.-P. Vanbremeersch

Round 2

Reviewer 1 Report

As it was written in the 1st revision, the paper presents a conceptual mathematical framework for enriching human-machine interactions to improve decision-making in a social organization. The Authors were asked to emphasize their contribution and clearly state the problem – this part has been done. 

The Authors also provided the reason why they did not refer to the existing solutions. The rest of the comments were also addressed. 

Overall, I am able to accept the paper in its present form.

Reviewer 3 Report

Authors did a good job explaining the novelty of the submitted paper and highlight the original contribution. Figures improved paper significantly. All of my comments on the original article have been addressed. Reviewer appreciates the effort of the authors, and the reviewer does not have any further comments.